# Vegetation-Dependent Response to Drought in Salt Marsh Ammonia-Oxidizer Communities

**DOI:** 10.3390/microorganisms8010009

**Published:** 2019-12-19

**Authors:** Jack K. Beltz, Hayley McMahon, Isis Torres Nunez, Anne E. Bernhard

**Affiliations:** 1School of Arts and Sciences Department of Biology, University of Pennsylvania, Philadelphia, PA 19014, USA; jkbeltz@sas.upenn.edu; 2Department of Ecology and Evolutionary Biology, University of Connecticut, Storrs, CT 06269, USA; hayley.mcmahon@uconn.edu; 3Department of Biology, Connecticut College, New London, CT 06320, USA; itorresn@conncoll.edu

**Keywords:** *amo*A, drought, disturbance, salt marsh, nitrifiers

## Abstract

We investigated the impacts of drought on ammonia-oxidizing archaea (AOA) and bacteria (AOB) in a salt marsh and compared the response to the total bacterial community. We analyzed abundance and community composition of *amo*A genes by QPCR and TRFLP, respectively, in three vegetation zones in 2014 (pre-drought), 2016 (drought), and 2017 (post-drought), and analyzed bacterial 16S rRNA genes by QPCR, TRFLP, and MiSeq analyses. AOA and AOB abundance in the *Spartina patens* zone increased significantly in 2016, while abundance decreased in the tall *S. alterniflora* zone, and showed little change in the short *S. alterniflora* zone. Total bacterial abundance declined annually in all vegetation zones. Significant shifts in community composition were detected in 2016 in two of the three vegetation zones for AOA and AOB, and in all three vegetation zones for total bacteria. Abundance and community composition of AOA and AOB returned to pre-drought conditions by 2017, while bacterial abundance continued to decline, suggesting that nitrifiers may be more resilient to drought than other bacterial communities. Finding vegetation-specific drought responses among *N*-cycling microbes may have broad implications for changes in N availability and marsh productivity, particularly if vegetation patterns continue to shift as predicted due to sea level rise.

## 1. Introduction

Nitrification, the sequential oxidation of ammonia to nitrate, is a critical process in salt marshes that controls the fate of nitrogen and impacts primary productivity in the marsh. However, we know very little about how changes in precipitation brought about by climate change, especially periods of drought, might impact the organisms that carry out nitrification. The first step of nitrification, the oxidation of ammonia to nitrite, is carried out by a suite of bacteria and archaea carrying the ammonia monooxygenase gene. Ammonia-oxidizing populations in estuarine systems have been well described in terms of abundance and community composition [1], but what regulates their activity and abundance in response to disturbance is less well characterized. In many cases, salinity has been shown to be an important factor regulating both ammonia-oxidizing archaea (AOA) [2] and ammonia-oxidizing bacteria (AOB) [3,4], while other studies have found marsh elevation and dominant vegetation to be important [5]. Drought is likely to lead to significant changes in salinity within the marsh, and the level of impact may be dependent on tidal elevation as well as the type of vegetation.

Although AOA and AOB carry out the same process, using a similar enzyme, recent studies have reported differences in how AOA and AOB respond to disturbance, with AOA appearing to be more resistant or resilient compared with AOB. For example, Peng et al. [6] found stronger responses to long-term fertilization in AOB communities compared with AOA in a New England salt marsh, and similarly, Bernhard et al. [7] found greater changes in AOB communities compared with AOA due to salt marsh impoundment and subsequent restoration. Others have reported increased sensitivity of AOA to drought in terrestrial systems [8,9]. However, in riparian systems, AOA were found to be more resilient compared with AOB after flooding disturbance [10]. And, Urakawa et al. [11] found AOA to be more sensitive to oiling compared with AOB in a laboratory study of cultured isolates. Thus, many questions remain regarding the factors that govern the health and productivity of these organisms and their response to disturbance in coastal systems, such as drought.

Drought conditions can have complex and variable impacts on salt marsh ecosystems. Periods of decreased rainfall and subsequent rewetting associated with drought have induced transient and nonuniform shifts in salt marsh vegetation composition [12], as well as significant marsh dieback events [13]. Results of previous research on the microbial response to drought have varied depending on habitat type, microbial community under study, and the drought duration [14,15]. Sediment waterlogging and oxygen depletion caused shifts in abundance and composition of nitrogen-fixing bacteria in a salt marsh rhizosphere habitat [14]. Riverbed drying and sediment cracking promoted growth of ammonia oxidizers, particularly AOA [16]. Additionally, the rewetting effect after a drought may also have significant effects, inducing a shift in microbial community composition with a similar magnitude as the drought event itself [17]. The impact of drought on ammonia-oxidizing communities in salt marshes, however, has not yet been studied. Since these communities are regulated by environmental conditions that are expected to be directly impacted by drought and rewetting, such as salinity and oxygen, we predict that ammonia-oxidizing communities may also be impacted by drought, which could lead to changes in how nitrogen is processed in the marsh and potentially, impact marsh productivity.

We have routinely collected samples in the Barn Island salt marshes since 2005 to study the ecology of ammonia oxidizers in the marsh, with particular emphasis on different vegetation zones. A major sampling was planned in 2016 to conduct a 10 year follow-up study to our earlier study in 2006 on recovery of N-cycling microbes after impoundment and restoration [7]. However, during our sampling, we noticed the marsh was much drier than normal (Appendix A), and subsequently learned that southeastern CT was experiencing a severe to extreme drought (National Oceanic and Atmospheric Administration) that lasted until spring 2017. Since we had collected samples in 2014 for a different study, we took advantage of our situation and added another sampling trip in late June 2017 to provide a post-drought comparison. In addition to assessing the impact of drought on AOA and AOB, we also compared the response to drought between ammonia oxidizers and the total bacterial community, to determine if the response was universal.

## 2. Materials and Methods

*Study Site.* Samples were collected in the Headquarters Marsh at the Wequetequock-Pawcatuck tidal marshes, part of the Barn Island Wildlife Management Area in Stonington, CT. The marshes are part of the Little Narragansett estuarine system, with a mean tidal range of 80 cm [18]. The marsh is fed by the Little Narragansett Bay and tributaries of the Pawcatuck river. Headquarters Marsh, located at lat/long 41.338829 N, 71.873048 W, is an eight-hectare section that borders the bay. More complete site descriptions and maps have been previously published [18,19]. The marsh is dominated by tall *Spartina alterniflora* (TSA), short *S. alterniflora* (SSA), and *S. patens* (SP), with areas of mixed vegetation including *Distichlis*, *Salicornia*, *Juncus*, and forbs.

*Sample Collection.* Sediment samples were collected at low tide on 24 June 2014, 13 and 19 July 2016, and 28 June 2017 using 6.5 cm diameter corers. Samples were collected from three different vegetation zones within the marsh: SP, SSA, and TSA. In 2014 and 2017, triplicate cores were collected from each of the three vegetation sites for a total of nine cores each year. In 2016, four cores were collected from SP, but only duplicate cores were taken at TSA and SSA since at the time of sampling the main focus was on SP sites. Samples were immediately stored in the cold and dark after collection until processing, less than 24 h later. Aliquots (0.5 g) were taken from the surface (0–2 cm) and stored at −80 °C. Porewater from each core was obtained from the remaining sediment by centrifugation (5000× *g* for 5 min) in 50 mL tubes with 0.45 μM cellulose acetate filter insert (Chrom Tech, Inc., Apple Valley, MN, USA). Porewater salinity was measured from each core using a hand-held refractometer. pH was measured with a pH meter or with pH paper depending on volume of porewater available. Ammonium was measured using the phenol-hypochlorite method [20] adapted for microtiter plates. Nitrate (NO_3_^−^ + NO_2_^−^) was measured using enzymatic nitrate reduction (NECi, Lake Lindin, MI, USA), followed by colorimetric analysis of nitrite. We were unable to obtain porewater from the TSA cores in 2016 because they were too dry.

*DNA Extractions.* DNA was extracted from sediment samples according to the manufacturer’s instructions using the DNeasy kit (MoBio/Qiagen, Carlsbad CA, USA). Two different editions of this kit were used across the study, and direct comparison of the two kits yielded no significant difference in DNA yield. The only modification made to the manufacturer’s protocol was an extended initial centrifugation time from 30 s to 5 min to ensure more complete separation of the supernatant and sediment particles. DNA was stored at −80 °C, and subsequent aliquots were stored at −20 °C. DNA concentration and purity were quantified using a NanoDrop Lite spectrophotometer (Thermo-Fischer Scientific, Waltham MA, USA). Samples with a 260:280 ratio below 1.5 were excluded, and DNA was extracted from replicate aliquots for those samples. Optimal dilutions of DNA for QPCR and sequencing were empirically determined by analyzing amplification from serial dilutions and selecting the dilution that minimized inhibition. For all samples, 1:10 dilutions were determined to be optimal and were used for all QPCR, TRFLP, and sequencing analyses.

*Quantitative PCR.* Betaproteobacterial *amo*A genes were quantified using primers amoA-1F and an equal mixture of amoA-2R/amoA-2RTC [21,22]. The mixture of reverse primers was empirically determined to yield more specific products compared with reactions using only one or the other reverse primer. All reactions were run in an iCycler or CFX (BioRad) using iQ SYBR Green I master mix (BioRad) with previously published protocols [5]. Archaeal *amo*A reactions were run using ArchAmoAF and ArchAmoAR primers [23], with one minute initial denaturation at 95 °C, followed by 50 cycles of 95 °C for 5 s, at 54 °C for 20 s, and 72 °C for 20 s. Bacterial 16S rRNA genes were amplified using the forward primer (GM3) and reverse primer (338R) as previously described [19]. Melt curve analysis was also performed after each experimental run to confirm product specificity. Sample amplification was compared with a standard curve generated in each experimental run using five standards generated from plasmids containing cloned genes ranging in concentration from 1 pg/μL to 0.0001 pg/μL for *amo*A genes and 1 ng/uL to 0.0001 ng/uL for 16S rRNA genes. Only data from runs with efficiencies greater than 85% and standard r-squared values of 99% ± 2% were used.

*T-RFLP Analysis.* Community composition for each sample was determined using terminal restriction fragment length polymorphism (TRFLP) for the betaproteobacterial and archaeal *amo*A genes as well as the bacterial 16S rRNA gene. We included TRFLP for bacterial 16S rRNA genes even though we also analyzed DNA sequences for these communities, so that we could make a direct comparison with the nitrifier communities using the same method. Previous studies of *amo*A genes in our lab using both TRFLP and Sanger sequencing have shown strong correlations between OTU distribution and TRFLP patterns for both AOA (*r* = 0.93) and AOB (*r* = 0.83) [24]. The betaproteobacterial *amo*A gene was amplified using PCR, with the fluorescently labeled (6-FAM) forward primer *amo*A-1F and an equal ratio of two reverse primers, *amo*A-2R and amoA-2RTC. The PCR and TRFLP protocols were carried out according to Bernhard et al. [7], using *Aci*I (NEB, Ipswich, MA). PCR and TRFLP of archaeal *amo*A and bacterial 16S rRNA genes were performed as previously described [24], using *Aci*I (archaea) and *Msp*I (bacteria). Samples were analyzed on an Applied Bio Systems 3730xl DNA Analyzer at the Biotechnology Resource Center at Cornell University (http://cores.lifescien ces.cornell.edu/brcinfo/). Terminal restriction fragment (TRF) length and relative abundances were estimated using GeneMarker software, v.1.4 (SoftGenetics, State College, PA, USA).

*Bacterial 16S rRNA MiSeq Analysis.* Samples from each vegetation zone and year were selected for Illumina MiSeq Next-Gen sequencing, with nine samples from 2014 and 2017 and six samples from 2016. Samples were selected based on DNA concentration and previous amplification success. Aliquots of selected samples were sent to the University of Connecticut Microbial Analysis, Resources and Service (https://mars.uconn.edu/)(Storrs, CT) for sequencing of the V4 region of the 16S ribosomal RNA gene using bacterial-specific primers (515F and 806R), generating paired 250-nucleotide reads. Paired-end reads were assembled into contigs and screened to remove low-quality reads and chimeras using mothur v1.35.9 (MiSeq_SOP 6/18) [25]. After filtering, we had a total of 577,029 sequences, with an average of 23,081 reads per sample. Data were rarefied to the lowest number of sequences per sample (11,712) for downstream analyses. OTUs were determined using 97% similarity. Sequences were assigned to the closest genus with an 80% confidence threshold using the mothur-formatted version of the Ribosomal Database Project training set (v.9). Analysis of molecular variance (AMOVA), homogeneity of molecular variance (HOMOVA), phylotyping (using the Silva database), rarefaction curve analysis, as well as two-dimensional (2D) non-metric multidimensional scaling (NMDS) ordination and lefse analysis [26] were all done using mothur. OTU abundance was determined for the non-drought years by averaging the read counts from 2014 and 2017 samples. Graphical analysis of OTU NMDS ordination and heatmap construction were created in R (vegan, plotly, heatmap.2), using abundance of OTUs calculated using the total sequence reads from 2016 for drought or the average sequence reads from 2014 and 2017 for non-drought. The sequences obtained were deposited into the GenBank database under accession #PRJNA518080.

*Statistical Analysis.* Differences in abundance (Q-PCR) between groups (sample years and vegetation zones) of samples were determined by t-tests and analysis of variance using R (v3.5.1). Patterns in community composition data were analyzed using non-metric multidimensional scaling (NMDS) [27] analyses in PC-Ord v1.6 [28]. Details of data analysis and specific procedures for NMDS ordination in PC-Ord can be found in Peng et al. [6]. Multi-response permutation procedure (MRPP), a nonparametric statistical test, was used to test for differences between drought and non-drought years, sample years, and vegetation zones. MRPP is an analysis of similarity and provides a measure of the effect (*p* value) when testing for differences between two or more groups [29].

## 3. Results

### 3.1. Sediment Conditions

Southeastern Connecticut experienced a drought beginning in July−August 2015, reaching the status of extreme drought in September 2016, based on the Palmer Drought Severity Index (PDSI) reported by NOAA (https://www7.ncdc.noaa.gov/CDO/CDODivisionalSelect.jsp#) (Appendix A). Not surprisingly, there was a significant decrease in rainfall one month prior to sampling in 2016 (Figure 1). This was mirrored by a significant increase in sediment salinity in the marsh during sampling in 2016. In 2017, there was an increase in rainfall, matched by an expected decrease in sediment salinity.

In 2016, porewater nitrate and ammonium were below detection in all samples, while in 2017, we detected nitrate in all samples and ammonium at all sites except SP (Table 1). Unfortunately, we do not have porewater nutrients for samples collected in 2014. There was a significant decrease in soil moisture at TSA during the drought year (*p* = 0.01), and although soil moisture increased by about 50% in 2017, it had not returned to pre-drought levels. At SP, there was also a decrease in soil moisture, but the differences were not significant (*p* = 0.19), and at SSA, there was no change in soil moisture during the drought.

### 3.2. Gene Abundance

AOA and AOB had similar abundance patterns within each vegetation zone, but the direction of the pattern differed between zones (Figure 2). In TSA, both AOA and AOB showed a significant decrease in abundance during the drought (2016). In SP samples, however, an inverse pattern was observed with a significant increase in abundance during the drought. The pattern in SP samples was also much stronger than in other vegetation zones, with AOB increasing by 4 orders of magnitude and exceeding AOA abundance, while AOA increased by 2 orders of magnitude. Abundance in SSA showed no significant differences in abundance for either AOA or AOB.

Since we detected significant effects of drought on AOA and AOB abundance, we were curious how this compared to the total bacterial community. To address this question, we quantified bacterial 16S rRNA genes. Unlike AOA and AOB, total bacterial abundance decreased in 2016 in all three vegetation zones, and was significantly lower in TSA and SSA, and continued to decrease in 2017. Bacterial 16S rRNA abundance patterns were similar in SP, but were not significant.

AOA and AOB abundances were positively correlated (*r* = 0.53, *p* < 0.05), but neither had significant correlations with environmental variables (Appendix A). The total bacterial community, on the other hand, was positively correlated with precipitation (*r* = 0.82, *p* < 0.05), while having a negative relationship to salinity (*r* = −0.52, *p* < 0.05).

### 3.3. Community Composition

Based on TRFLP analysis, total bacteria, AOA, and AOB community composition all differed significantly among the three vegetation zones. Additionally, AOA and AOB were significantly impacted by drought in two of the vegetation zones (Figure 3). AOA community composition changed significantly during the drought in TSA and SSA, while drought effects were detected in TSA and SP for AOB communities. The total bacterial community, however, shifted significantly in all three vegetation zones. In all cases where there was a significant drought effect, communities in 2017 were not significantly different from those in 2014.

In all samples, AOA communities were dominated by TRF170 and TRF296 (Appendix A), which represent sequences related to *Nitrosopumilus maritimus* (as described in [24]). AOA TRF83 increased during the drought year in all vegetation zones. Conversely, the dominant AOB populations varied with vegetation and drought (Appendix A), with most of the dominant TRFs (TRF127, 130, and 278) representing sequences that correspond to *Nitrosospira*-like sequences (as described in [6,24]). In 2017, TRF196 increased in SSA and SP plots, and represents sequences that correspond to *Nitrosomonas*, which is consistent with patterns observed in sequence abundance analysis of nitrifier OTUs (Appendix A).

### 3.4. MiSeq Analysis of 16S rRNA Genes

We detected 15,575 distinct OTUs (with singletons removed) from all samples. The majority of these OTUs were unique to TSA communities (Figure 4A), with only 11% shared among all vegetation zones. Communities in SP and TSA zones were the most similar, and TSA/SSA the most distinct. PCA ordination of the samples based on OTU relative abundance (Figure 4B) revealed significantly different communities in each vegetation zone and a significant drought effect within each zone. Bacterial communities were significantly more diverse (Inverse Simpson’s Index) during the drought year in SSA and SP zones (Table 2), while the opposite, but nonsignificant, pattern was observed in the TSA zone.

We identified the 20 most abundant OTUs in each vegetation zone and compared changes in relative abundance between drought and non-drought years (Figure 5). We found distinct patterns among vegetation zones, but in all cases, there was a general pattern of decreased abundance of the dominant OTUs in the drought year samples. SSA had the most changes in abundant taxa during the drought, with ten of the twenty most abundant OTUs showing significant declines during the drought year. Only three or four OTUs declined significantly in abundance in TSA and SP, respectively. Unlike in SSA, there was only one taxon in TSA and SP that increased in abundance during the drought.

## 4. Discussion

Drought frequency and duration are predicted to continue to increase around the world as climate change remains a pressing problem [30]. Understanding how changes in precipitation impact nitrogen-cycling communities is imperative to track broader impacts of climate change on salt marshes. The drought in Southeastern Connecticut in 2016 altered the sediment chemistry of the Barn Island salt marshes, which would likely cascade into biotic systems, including critical *N*-cycling microbes that promote salt marsh productivity. In this study, we detected a significant impact of drought on ammonia-oxidizing communities, as well as the bacterial community as a whole. The drought response varied by microbial community and vegetation zone, with AOA and AOB showing strong agreement and potentially, a higher level of resilience relative to the total bacterial community.

*Nitrifier Abundance*. The similar direction of drought response of AOA and AOB abundance within each vegetation zone suggests similar mechanisms may regulate both groups in response to drought and rewetting. In studies of AOA and AOB during drought conditions, there is a general pattern of differential responses of the two groups. For example, Fuchslueger et al. [17] reported decreased AOA abundance, but stable AOB abundance, during drought in a managed meadow. Others have also reported differential responses in recovery times [31], with AOA appearing to be more sensitive to drought than AOB [8]. While both AOA and AOB abundance in our study responded in a similar direction within each vegetation zone, the magnitude of the impact varied, with AOA having a stronger response compared with AOB in TSA, and AOB responding more strongly than AOA in SP. The differences in magnitude of change suggest that the two groups have different sensitivities to drought conditions that may also be influenced by interactions with different vegetation and/or soil conditions. Gleeson and colleagues [32] reported higher AOA:AOB ratios in the wettest soils in a study of semi-arid soils. Thion and Prosser [8] suggest that differences in responses to drought between AOA and AOB are related to changes in ammonium concentrations, but Bello et al. [9] suggest that the differences may actually be due to different sensitivities to osmotic stress, with AOA being more sensitive. Differential responses of AOA and AOB to drought may have important consequences for the ecosystem since AOB contribute more of the greenhouse gas, N_2_O, compared with AOA during nitrification [33].

The opposite responses of AOA and AOB abundance observed between TSA and SP during the drought year may suggest unique sediment conditions in these zones or distinct interactions with the different vegetation species. In a study of AOA and AOB abundance along a salt marsh chronosequence, Wang et al. [34] found that plant host had a stronger influence on nitrifier abundance than soil conditions, and Delgado-Baquerizo et al. (36) argue that abundance of AOA and AOB is a reflection of vegetation that modulates the local environment. In salt marshes, the vegetation zone is determined by degree of tidal flooding, which also has a strong influence on soil conditions. Therefore, it is difficult to know if the different responses in TSA and SP zones are being promoted by the vegetation or by the soil conditions brought about by tidal flooding.

Increased AOB gene abundance has been previously documented under extended drought conditions in a variety of ecosystems [17,35], and Palomo et al. [36] found increased nitrification in sediments shifting from anoxic to more oxic conditions during drought. However, Stark and Firestone [37] reported decreased nitrification activity during drought in soils. We think it is likely that increased abundance of AOA and AOB in the SP sites may be related to oxygen availability. Sediment drying during the drought in the marsh promoted significant crack formation in the surface layer (Appendix A), likely leading to increased oxygen penetration and potential growth of these aerobic microbes. In a study of a freshwater system, riverbed drying and cracking was also associated with an increase in nitrifier abundance, which was attributed to greater oxygen availability [16]. Bello et al. [9] found AOA to be more sensitive to changes in osmotic stress in a soil microcosm experiment compared with AOB, suggesting they may also be more sensitive to drought, particularly in salt marshes, where increased salinity would likely induce more osmotic stress for the organisms.

Interestingly, finding higher abundances of both AOA and AOB in TSA sites compared with SP sites in non-drought years varies from previous reports in these same sites from 10 years prior [5], where abundance was higher in SP sites. June and July 2006 were considered very moist spells in coastal CT (based on PDSI data from NOAA), which may contribute to the differences in abundance. It is also possible that the change in abundance patterns could reflect changes due to sea level rise [38]. However, AOB abundance in the SP zone during the drought in our study are the highest values for AOB in any habitat that we could find. Although AOA frequently outnumber AOB in studies from a variety of habitats, including salt marshes (e.g., [5,6]), our data indicate that under certain conditions, AOB outcompete AOA. In a meta-analysis including data from seven published studies in estuaries or salt marshes where abundance of both AOA and AOB were measured [2,4,5,6,39,40,41], we found AOA were more abundant than AOB in only 51% of the samples, suggesting that the competitive dominant varies in these dynamic estuarine systems.

The cause of the decrease in abundance in the low marsh TSA zone is unclear. Studies have shown that salinity is often negatively correlated with nitrifier abundance, and although we did measure higher salinity overall, we were unable to measure salinity at TSA sites during the drought due to a lack of extractable porewater. Soil moisture at this site was much lower overall even in non-drought years, and during the drought dropped more than 40%, while moisture in the other sites changed by only ~4% (SP) or not at all (SSA). Such dramatic changes in soil moisture would likely have a detrimental effect on microorganisms due to osmotic stress. It is also possible that the lack of porewater observed in TSA samples could equate with decreased nutrient availability due to decreased diffusion in dry soils [37]. Furthermore, Davis et al. [14] reported a decrease in nitrogen-fixing diazotroph populations in a salt marsh undergoing drought, which could lead to a decrease in available ammonium. Further characterization of other sediment chemistry parameters and microbial activity is necessary to determine what is driving the decreased nitrifier abundance at the TSA site.

*Nitrifier Community Composition*. The significant drought effect on community composition observed in two of three vegetation zones for AOA and AOB suggests that the conditions are selecting for drought-tolerant species. Disturbance has previously been associated with changes in AOB community composition in studies of marshes [6,42] and grasslands [17]. Because changes in AOB community composition have been linked to shifts in nitrification rates (e.g., [3]), finding drought-induced shifts in both AOA and AOB could indicate changes in ecosystem function during drought conditions. Finding a drought-associated shift from *Nitrosospira* 16S rRNA sequences to *Nitrosomonas* sequences in some sites would be expected to lead to changes in nitrification rates, since *Nitrosomonas* typically has higher growth rates relative to *Nitrosospira* [43].

There was a significant impact of drought on the AOB community composition in the SP zone, as well as on abundance. AOA community composition in the SP zone, however, showed no significant drought effect. Our inability to detect a drought effect in the AOA community may be due to the high variation in replicate samples during the drought year. Variability in communities can indicate a heightened capacity for adaptation or response to disturbance [19], and decreased community stability has been attributed to heightened resilience potential [44]. Regardless of significance, both AOA and AOB communities in the SP zone had strong shifts in the drought year, which corresponded with a concurrent increase in abundance. These compositional changes could similarly be associated with shifts in oxygen availability, as suggested by Davis et al. [14].

*Total Bacterial Abundance*. The decrease in total bacterial abundance during the drought year followed by a continued decline post-drought is distinct from the pattern observed in the ammonia-oxidizing communities, which showed signs of recovery by 2017. This may indicate that nitrifier communities are more resilient than other bacterial communities. A decline in microbial biomass during drought has been documented previously [45] and was attributed to changes in soil moisture and a lack of rhizospheric C and N availability. Sediment bacterial communities have also been negatively impacted by drought conditions, with patterns varying greatly among phyla [46]. Increases in long-term drought-related sediment aridity has been shown to decrease microbial abundance and diversity [47], suggesting a longer-lasting impact of drought or an additional secondary disturbance triggered from the post-drought rewetting. Rewetting itself has been shown to be a significant disturbance [48], and the magnitude of the rewetting effect in some cases is stronger than the drought effect [14]. Rewetting may have served as an additional disturbance, contributing to the continued decline observed in the total bacterial community. Differences in the recovery rate of nitrifier abundance compared with the total bacterial abundance suggest that disturbance, such as drought or rewetting, may differentially impact microbial populations, which may have implications on nutrient cycling in the marsh.

*Total Bacterial Community Composition*. The purpose of analyzing bacterial community composition was to determine if the response of the nitrifiers was similar to that of the larger bacterial community, not to systematically identify drought-tolerant or sensitive groups. Based on both TRFLP and sequence analysis, we showed that bacterial communities shifted during the drought, similar to nitrifiers, and also similar to nitrifiers, showed a shift back to communities indistinguishable from pre-drought communities in the summer following the most severe drought conditions. However, the drought effect on bacterial communities was broader than that on nitrifiers, with significant community shifts detected in all three vegetation zones, while for nitrifiers, communities in some vegetation zones were less affected by the drought, suggesting either a more resistant community or a protective effect of the vegetation on nitrifiers. Total bacterial community composition has been reported to be impacted by various types of disturbances [49,50], and drought specifically has been identified as a driver of community composition [48]. It is thought that salinity may play a pivotal role [51]. This supports our findings of significant increases in salinity during the drought that were accompanied by significant community shifts. Naylor et al. [15] has also established that the intensity of the bacterial compositional shift is directly correlated with drought intensity. Whether this is also true for nitrifiers specifically cannot be determined from our study. In addition to salinity, soil moisture has had the greatest impact on the composition of rhizospheric communities in soils [52,53], and both of these parameters are likely significantly altered during a drought.

## 5. Conclusions

Our results suggest a significant drought impact on the three microbial communities surveyed, and that the response varied between nitrifiers and the total bacterial community. The different post-drought responses by nitrifiers compared with total bacteria suggests a drought resilience in the nitrifier community that is absent in the bacterial community, and possible enhanced drought-tolerance by *Nitrosomonas* species. Since others have also reported increased resilience of nitrifiers in other habitats [14,17], it is possible that the resilience is a universal trait among nitrifiers. This resilience has been suggested to be an advantageous community trait, which minimizes *N*-cycling disruption in periods of drought [54]. Longer-term monitoring will be important to fully elucidate the effects of the drought. Additionally, in situ measurements of nitrifier activity during drought is a critical metric which would explain the contribution of physical responses such as dormancy, as well as gain a better understanding for functional resilience or compensation by these communities. A more complete understanding of how salt marsh *N*-cycling will change over time is imperative to understand how climate change will shape the productivity of these systems.

We acknowledge that our study is limited by having only one year of drought conditions, and we cannot rule out that the variation we see in the drought and non-drought years is simply normal annual variation in the marsh. However, we think the patterns we observed in the two non-drought years that were different from the drought year suggest that something caused the change. An additional complicating factor is the effect of sea level rise, which makes it difficult to compare data from 10 years prior in the marsh. We believe that, despite the unavoidable limitations of our study design, our study provides useful information about potential impacts of climate change and the responses of the microbial communities.

Finally, drought has been associated with the timing of acute marsh dieback events in several studies [13,55]. A disruption in nitrogen availability by shifts in the function or abundance of microbial communities, particularly nitrifiers, caused by drought conditions could be contributing to such events. The physical changes to sediment structure associated with droughts have been shown to impact vegetation recovery rate significantly as well [56]. The ability for a community to withstand or rebound from drought will be an important quality to predict the full impact of climate change in salt marshes.

## Figures and Tables

**Figure 1 microorganisms-08-00009-f001:**
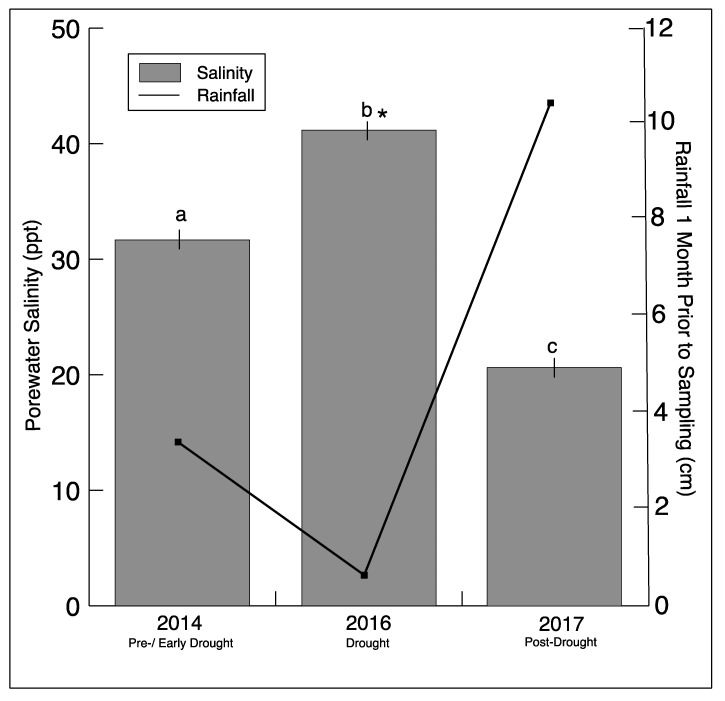
Rainfall totals one month prior to sampling (NOAA/AHPS, water.weather.gov/precip) for Stonington, CT, and average salinity (± SD)_from all sediment sample porewater for each sampling year. Significantly different (*p* < 0.05) salinity values for each year are indicated by different letters. The asterisk indicates a significant difference between drought and non-drought years.

**Figure 2 microorganisms-08-00009-f002:**
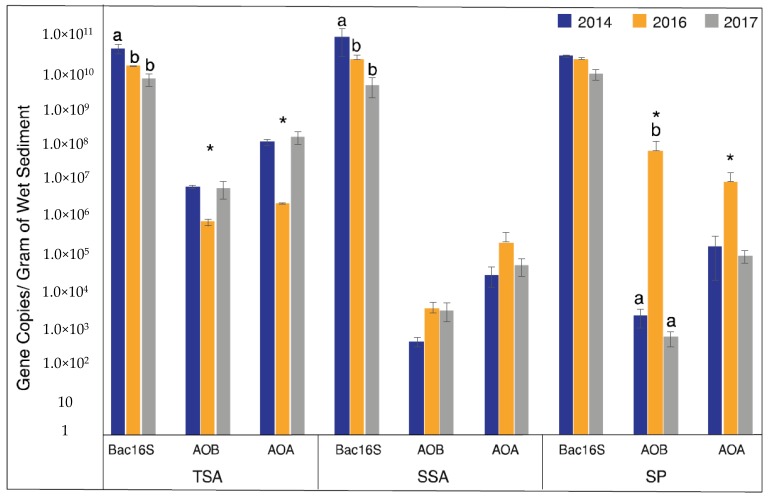
Mean gene abundance ± standard error over the three years and vegetation types. Different letters indicate significantly different values among sample years (ANOVA, *p* < 0.05), and asterisks indicate the drought year (2016) was significantly different from non-drought years (2014 and 2017 combined) (Welch t-test, *p* < 0.05).

**Figure 3 microorganisms-08-00009-f003:**
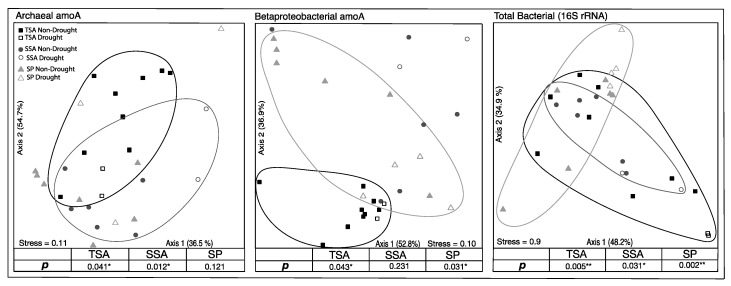
Non-metric multidimensional scaling ordination (NMDS) based on relative abundance of unique community terminal restriction fragments (TRFs). Circled points indicate community composition of drought year samples in that vegetation zone were significantly different from non-drought year samples (MRPP, adjusted *p* < 0.05). Percent variation explained by each axis is indicated, as well as the final stress on the ordination.

**Figure 4 microorganisms-08-00009-f004:**
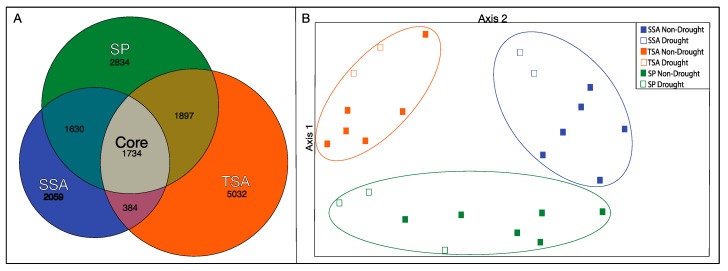
Euler diagram indicating the number of unique (excluding singletons) OTUs identified within each vegetation zone and with all combined (core) (Panel **A**). Area of each space represents relative proportion of total OTU diversity (R “eulerr”). Principal components analysis (PCA) of total bacterial communities grouped by vegetation zone, based on relative abundance of OTUs identified in MiSeq analysis (Panel **B**). Circled points indicate vegetation zones where there was a significant difference between non-drought and drought year samples (HOMOVA, *p* < 0.05).

**Figure 5 microorganisms-08-00009-f005:**
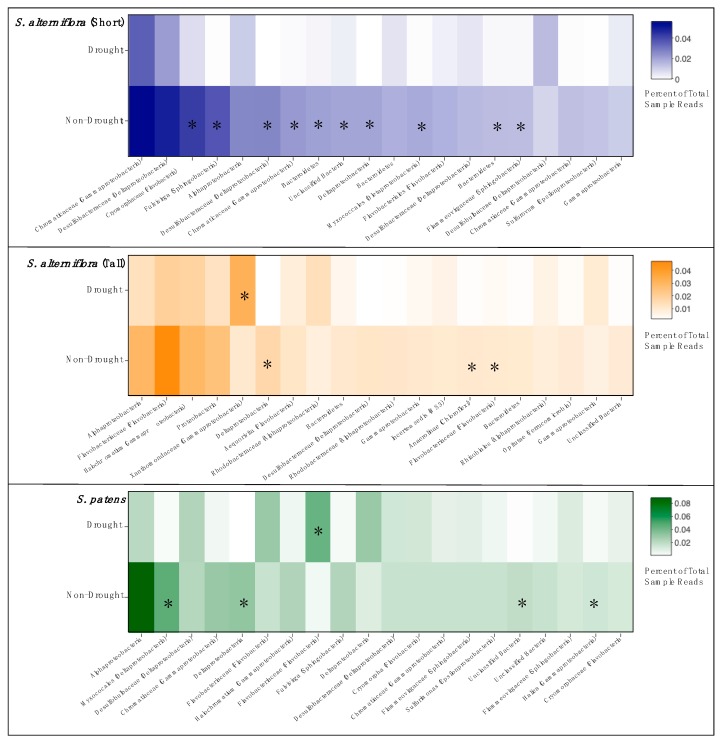
Relative abundance heatmaps of the top twenty most abundant OTUs from each vegetation zone, comparing drought year and non-drought years. OTU taxa are labeled with lowest identified classification and phyla parenthetically, if available. Asterisks indicate a significant difference in abundance between drought and non-drought years (*p* < 0.05, HOMOVA).

**Table 1 microorganisms-08-00009-t001:** Porewater sediment conditions in the three vegetation zones in 2014 (pre-drought), 2016 (during drought), and 2017 (post-drought). SP = Spartina patens, SSA= Short S. alterniflora, TSA = tall S. alterniflora. Different letters indicate significantly different values within each vegetation zone (site). For salinity, we could only test for significant differences between 2016 and 2017 since we have site salinity for 2014 rather than salinity from individual cores.

Site	Year	Salinity	Soil Moisture (%)	Nitrate (µm NO_3_^−^-N)	Ammonium (µM NH_4_^+^-N)
SP	2014	33.0	87.8 ± 0.8	-	-
	2016	42.2 ± 13.6 ^a^	83.3 ± 0.04	bd	bd
	2017	24.3 ± 3.8 ^b^	85.4 ±2.1	2.8 ± 4.4	bd
SSA	2014	34.0	87.8 ± 1.5	-	-
	2016	39.0 ± 2.8 ^a^	87.8 ± 0.0	bd	bd
	2017	20.3 ± 1.2 ^b^	88.8 ± 2.2	9.9 ± 3.3	0.1 ± 0.1
TSA	2014	28.0	62.3 ± 13.3 ^a^	-	-
	2016	na	20.5 ± 0.1 ^b^	na	bd
	2017	23.3 ± 2.9	47.8 ± 6.1^a^	23.7 ± 19.6	18.9 ± 10.9

bd = below detection; “-” indicates parameter was not measured; na = no porewater was available.

**Table 2 microorganisms-08-00009-t002:** Mean bacterial diversity (Inverse Simson index) values in each vegetation zone for each year based on Bacterial 16S rRNA Miseq analysis. Significantly different values among years within each vegetation zone are indicated by different letters.

*Inverse Simpsons*	2014	2016	2017
*ALL VEG*	253.5 ± 117.5 ^a^	332.3 ± 115.4 ^a^	236.2 ± 206.3 ^a^
*TSA*	370.2 ± 92.8 ^a^	356.1 ± 181.1 ^a^	479.9 ± 188.0 ^a^
*SSA*	158.3 ± 7.2 ^a^	233.5 ± 50.2 ^b^	129.7 ± 16.8 ^a^
*SP*	232.1 ± 108.9 ^a,b^	382.1 ± 90.6 ^b^	99.1 ± 18.3 ^a^

^a,b^ Different letters indicate significantly different values within each vegetation zone (site).

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
