# Peer review of "Vegetation-Dependent Response to Drought in Salt Marsh Ammonia-Oxidizer Communities"

_microorganisms, 2019, doi:10.3390/microorganisms8010009_

Round 1
Reviewer 1 Report
In this manuscript the authors described the effect of drought in salt marsh ammonia oxidizers communities. The microbial communities were characterized by qPCR, T-RFLP and 16S rRNA gene high-throughput sequencing. The results showed that both ammonia oxidizers and total bacteria were influenced by drought but ammonia oxidizers were more resilient. The manuscript is well written and well organized. Furthermore, the findings could be helpful to understand the effects of climate change on fragile ecosystems such as salt marsh. I have only few comments for the authors before the manuscript can be accepted.
Main issues:
The M&M section on the 16S rRNA gene sequencing (lines 140-154) should be improved. Where the samples amplified at different dilutions to check for PCR inhibition? Please report the primers sequences. Where the primers specific for Bacteria or also archaeal 16S was amplified? What is the expected length of the sequenced fragment? Which was the length of the reads? Was a paired-end sequencing performed? How were the obtained reads filtered? Which the similarity threshold used for OTU clustering? Which is the number of reads (total and average per sample) before and after filtering? Which is the confidence of the taxonomic assignment using Silva? Was a rarefaction analysis performed to ensure that the sequencing depth is enough? If not, I suggest to perform the rarefaction analysis and to rarefy the samples to the same number of sequences. I understand that the authors did not want to calculate average values for the abundance of the OTUs in 2016 since they have only two samples for two sites (lines 149-153), however I think that the same approach should be used for all the samples (i.e. average abundances or cumulative abundance for all the samples should be calculated). The qPCR results were expresses as Gene copy number/Gram of wet sediment. I was wondering if it is better to report the data as Gene copy number/Gram of dry sediment, since the results could be influenced by the different moisture of the samples. Why the most abundant AOA and AOB that were found by T-RFLP were almost not detected by 16S rRNA gene sequencing (also check lines 397-398 and Figure S5)? Is it possible that the terminal restriction fragments belong to different microorganisms?
Minor issues:
Line 36. I think that “reviewed by” could be removed. Similarly check also further in the manuscript, e.g. lines 225, 228, 323. Line 74. This should be the first time that the abbreviation CT is used. It should be explained. Lines 113-114 and lines 132-133. Can a reference for this approach (i.e. mixing of two reverse primers) be added? Lines 134-135. Which restriction enzyme was used? Lines 165-166. This sentence seems to repeat similar information twice (i.e. drought was experienced in 2016 and the status of severe drought was experienced in 2016). Figure 1. Can the error bars be reported? Line 174. Please check how p-values are reported in the manuscript (e.g. at lines 179, 181, 197, 198 and further) and be consistent. Table 1. Significant differences should be reported in the table. Lines 202-204. Please add significant differences also in Figure 2. 324-327. Please add the references. Lines 254-257. How were significant different abundances calculated since for the samples collected in 2016 total sequences and not average sequences were considered. Figure S5. The relative abundance should be reported instead than the reads number.
Author Response
In this manuscript the authors described the effect of drought in salt marsh ammonia oxidizers communities. The microbial communities were characterized by qPCR, T-RFLP and 16S rRNA gene high-throughput sequencing. The results showed that both ammonia oxidizers and total bacteria were influenced by drought but ammonia oxidizers were more resilient. The manuscript is well written and well organized. Furthermore, the findings could be helpful to understand the effects of climate change on fragile ecosystems such as salt marsh. I have only few comments for the authors before the manuscript can be accepted.
Main issues:
The M&M section on the 16S rRNA gene sequencing (lines 140-154) should be improved. We have revised this section to include the requested information.
Where the samples amplified at different dilutions to check for PCR inhibition?
Please report the primers sequences.
Where the primers specific for Bacteria or also archaeal 16S was amplified?
What is the expected length of the sequenced fragment?
Which was the length of the reads?
Was a paired-end sequencing performed?
How were the obtained reads filtered?
Which the similarity threshold used for OTU clustering?
Which is the number of reads (total and average per sample) before and after filtering?
Which is the confidence of the taxonomic assignment using Silva?
Was a rarefaction analysis performed to ensure that the sequencing depth is enough? If not, I suggest to perform the rarefaction analysis and to rarefy the samples to the same number of sequences. Yes, we performed rarefaction, and have added this information to the methods section.
I understand that the authors did not want to calculate average values for the abundance of the OTUs in 2016 since they have only two samples for two sites (lines 149-153), however I think that the same approach should be used for all the samples (i.e. average abundances or cumulative abundance for all the samples should be calculated). We have corrected this. We did use number of sequence reads for all the samples, but it wasn't explained correctly.
The qPCR results were expresses as Gene copy number/Gram of wet sediment. I was wondering if it is better to report the data as Gene copy number/Gram of dry sediment, since the results could be influenced by the different moisture of the samples. We have calculated the data using both methods, and the overall patterns are similar, but we chose to use the g wet weight since this unit is more commonly reported, so cross-study comparisons can be made.
Why the most abundant AOA and AOB that were found by T-RFLP were almost not detected by 16S rRNA gene sequencing (also check lines 397-398 and Figure S5)? Is it possible that the terminal restriction fragments belong to different microorganisms? AOA would not be detected in the 16S data set since they are archaea and our 16S data are bacterial-specific. For AOB, we believe that some of the TRFs are represented by the unclassified Nitrosomonadaceae, shown in Figure S5. The dominant TRFs are classified as Nitrosospira-like since there are no cultured representatives from this clade of sequences, which is why they would appear as unclassified. In lines 397-398, we meant that the recovery from the drought was absent from the bacterial community as a whole, not the actual sequences. We have revised this text to be more clear.
Minor issues:
Line 36. I think that “reviewed by” could be removed. Similarly check also further in the manuscript, e.g. lines 225, 228, 323. We have made these revisions.
Line 74. This should be the first time that the abbreviation CT is used. It should be explained. We have explained the abbreviation here.
Lines 113-114 and lines 132-133. Can a reference for this approach (i.e. mixing of two reverse primers) be added? We found that by adding more of one of the degeneracies to the mix gave us better results. We don't know of a reference that uses this particular approach, but have added a sentence to the methods explaining why we used this mixture.
Lines 134-135. Which restriction enzyme was used? We have added this information to the methods.
Lines 165-166. This sentence seems to repeat similar information twice (i.e. drought was experienced in 2016 and the status of severe drought was experienced in 2016). We have revised the text accordingly.
Figure 1. Can the error bars be reported? There are no error bars for rainfall since the values reported are the total rainfall for the month prior to sampling for each year. There are error bars for SD in the figure, but perhaps they didn't show up in the file?
Line 174. Please check how p-values are reported in the manuscript (e.g. at lines 179, 181, 197, 198 and further) and be consistent. We have made these corrections.
Table 1. Significant differences should be reported in the table. We have added this information.
Lines 202-204. Please add significant differences also in Figure 2. We have included these on the original figure, and wonder if they didn't translate in the downloaded version? We have different letters to indicate significant differences between years, and asterisks indicate significant differences between drought and non-drought years. We will confirm that these appear in the final proofs if the paper is accepted.
324-327. Please add the references. We have added these references.
Lines 254-257. How were significant different abundances calculated since for the samples collected in 2016 total sequences and not average sequences were considered. The average of 2014 and 2017 (the two non-drought years) was used to compare to the drought year. For yearly comparisons, no averages were used. We don't know of any other way to compare drought to non-drought specifically (as opposed to yearly comparisons).
Figure S5. The relative abundance should be reported instead than the reads number. We have remade the figure using relative abundance.
Reviewer 2 Report
GENERAL COMMENTS
This paper examines changes in time of overall community structure and abundance of nitrifiers (specifically ammonia oxidizers) in salt marsh soils. Because one of the years was substantially drier (lower precipitation) than the year preceding it or following it, the authors conclude that changes that occurred from year one to year two were manifestations of drought. They then related the observations to climate change.
The basic approach is reasonable, and I was glad to see that the authors addressed at least one important guild, as opposed to providing long lists of names of taxa recovered (which more often than not fail to inform the reader about possible functional changes that may or may not accompany any shifts in taxa present.
However, there is a serious problem with the experimental design that was used. There is no measure of the typical interannual variation associated with the marsh soils examined. Three years were used as an experimental period, and, largely, each was different from the other. I would think that a few sequential “normal” years would need to be included to see if the “drought” year falls outside that “normal variation.” It is still a problem, however, to have only one year upon which to base the “drought” effect.
SPECIFIC COMMENTS
The very 1st sentence of the body of the paper (line 30 – 33) is both awkward and ambiguous (and it is incorrectly punctuated). As such, it does not do a very good job of introducing the material to follow. The authors might consider replacing it with the following passage, or something similar:“Nitrification, the sequential oxidation of ammonia to nitrate, is a critical process in salt marshes, and it controls the fate of nitrogen and influences primary productivity in the marsh. However, we know very little about how changes in precipitation brought about by climate change, especially periods of drought, might affect the organisms that carry out nitrification.”
36 – 37: Another awkward sentence. Suggest the second clause be, “but what regulates their activity, abundance, and response to disturbance is less well-characterized.”l. 81: additional location information is required here. Latitude and longitude should definitely be included, and a map, while not mandatory, would be useful in helping the reader understand exactly where this site is. Furthermore, there is comment about an impoundment in the marsh, and although there is reference to an earlier paper, the size and location of that impoundment and its relationship to the current study are not given here. Because there are no details about the location of the impoundment, Headquarters Marsh cannot be located on any map that I have managed to uncover. Finally in line 83, I think the authors mean to say that the marsh is fed by the Little Narragansett Bay, for I can find no connection with Narragansett Bay proper. l. 87: Forbes should be forbs. I stopped marking editorial needs because there were so many. Any revision of this manuscript should be done with the aid of a strong proofreader.
Author Response
This paper examines changes in time of overall community structure and abundance of nitrifiers (specifically ammonia oxidizers) in salt marsh soils. Because one of the years was substantially drier (lower precipitation) than the year preceding it or following it, the authors conclude that changes that occurred from year one to year two were manifestations of drought. They then related the observations to climate change.
The basic approach is reasonable, and I was glad to see that the authors addressed at least one important guild, as opposed to providing long lists of names of taxa recovered (which more often than not fail to inform the reader about possible functional changes that may or may not accompany any shifts in taxa present.
However, there is a serious problem with the experimental design that was used. There is no measure of the typical interannual variation associated with the marsh soils examined. Three years were used as an experimental period, and, largely, each was different from the other. I would think that a few sequential “normal” years would need to be included to see if the “drought” year falls outside that “normal variation.” It is still a problem, however, to have only one year upon which to base the “drought” effect.
We appreciate the concerns of the reviewer about the experimental design, and agree that the story would be stronger if we had additional drought years. Unfortunately, we do not have additional years of drought conditions to include in the manuscript. We agree that the variation we see could be within the normal year-to-year variation of the marsh. However, we see a fairly strong and consistent pattern in the two non-drought years that is different from the drought year, suggesting that something happened in 2016 (the drought year) to cause the variation. Perhaps there are other normally-occurring conditions that would lead to these differences, and we cannot rule that out. We have added language to the discussion to acknowledge this limitation in our study. An additional complicating factor in the marsh is the effect of sea-level rise, which makes it difficult to compare data from 10 years prior, since we cannot account for these effects in the study design. We feel that, despite the limitations of the study design, the data provide useful information about potential impacts of climate change on marsh nitrifiers.
SPECIFIC COMMENTS
The very 1st sentence of the body of the paper (line 30 – 33) is both awkward and ambiguous (and it is incorrectly punctuated). As such, it does not do a very good job of introducing the material to follow. The authors might consider replacing it with the following passage, or something similar:
“Nitrification, the sequential oxidation of ammonia to nitrate, is a critical process in salt marshes, and it controls the fate of nitrogen and influences primary productivity in the marsh. However, we know very little about how changes in precipitation brought about by climate change, especially periods of drought, might affect the organisms that carry out nitrification.”
We have modified the sentence as suggested.
36 – 37: Another awkward sentence. Suggest the second clause be, “but what regulates their activity, abundance, and response to disturbance is less well-characterized.”
We have modified this sentence.
81: additional location information is required here. Latitude and longitude should definitely be included, and a map, while not mandatory, would be useful in helping the reader understand exactly where this site is. We have added lat/long coordinates. The site (Headquarters marsh) has been described (including maps) in several previous papers, so we did not feel it was necessary to provide a map for this study, and have added these references to the manuscript clearly indicating that the maps are available there.
Furthermore, there is comment about an impoundment in the marsh, and although there is reference to an earlier paper, the size and location of that impoundment and its relationship to the current study are not given here. Because there are no details about the location of the impoundment, Headquarters Marsh cannot be located on any map that I have managed to uncover. We have removed the reference to the impoundment since it is irrelevant to the current study.
Finally in line 83, I think the authors mean to say that the marsh is fed by the Little Narragansett Bay, for I can find no connection with Narragansett Bay proper. We have made this correction.
Round 2
Reviewer 1 Report
I thank the authors for addressing my comments.
Reviewer 2 Report
The strength of the conclusion of drought response is too great given the lack of direct evidence. The words added in the discussion are helpful, but they do not mitigate the title which declares that the observations are induced by a short-term drought. There are experiments that could be done (say with soil/sediment cores) that could demonstrate definitively whether or not the observed effect is "caused" by drought conditions.
Perhaps a better approach would be to change the focus of the manuscript to interannual variation (the literature is lacking in studies of interannual dynamics in guilds), then point out that the anomalous year was especially dry.